# RNA-Seq Analysis Identifies Transcription Factors Involved in Anthocyanin Biosynthesis of ‘Red Zaosu’ Pear Peel and Functional Study of *PpPIF8*

**DOI:** 10.3390/ijms23094798

**Published:** 2022-04-27

**Authors:** Zhenyu Ma, Chuangqi Wei, Yudou Cheng, Zhonglin Shang, Xiulin Guo, Junfeng Guan

**Affiliations:** 1College of Life Science, Hebei Normal University, Shijiazhuang 050024, China; mazhenyuqqtt@163.com (Z.M.); shangzhonglin@hebtu.edu.cn (Z.S.); 2Institute of Biotechnology and Food Science, Hebei Academy of Agriculture and Forestry Sciences, Shijiazhuang 050051, China; weichuangqi@163.com (C.W.); chengyudouyn@163.com (Y.C.); myhf2002@163.com (X.G.); 3Plant Genetic Engineering Center of Hebei Province, Shijiazhuang 050051, China

**Keywords:** *Pyrus*, anthocyanin, RNA-Seq, transcription factors, WGCNA, PpPIF8

## Abstract

Red-skinned pears are favored by people for their attractive appearance and abundance of anthocyanins. However, the molecular basis of anthocyanin biosynthesis in red pears remains elusive. Here, a comprehensive transcriptome analysis was conducted to explore the potential regulatory mechanism of anthocyanin biosynthesis in ‘Red Zaosu’ pear (*Pyrus pyrifolia* × *Pyrus communis*). Gene co-expression analysis and transcription factor mining identified 263 transcription factors, which accounted for 6.59% of the total number of transcription factors in the pear genome in two gene modules that are highly correlated with anthocyanin biosynthesis. Clustering, gene network modeling with STRING-DB, and local motif enrichment analysis (CentriMo) analysis suggested that PpPIF8 may play a role in anthocyanin biosynthesis. Furthermore, eight PIFs were identified in the pear genome, of which only *PpPIF8* was rapidly induced by light. Functional studies showed that PpPIF8 localizes in the nucleus and is preferentially expressed in the tissue of higher levels of anthocyanin. The overexpression of PpPIF8 in pear peel and pear calli promotes anthocyanin biosynthesis and upregulates the expression of anthocyanin biosynthesis genes. Yeast-one hybrid and transgenic analyses indicated that PpPIF8 binds to the *PpCHS* promoter to induce *PpCHS* expression. The positive effect of PpPIF8 on anthocyanin biosynthesis is different from previously identified negative regulators of PyPIF5 and MdPIF7 in pear and apple. Taken together, our data not only provide a comprehensive view of transcription events during the coloration of pear peel, but also resolved the regulatory role of PpPIF8 in the anthocyanin biosynthesis pathway.

## 1. Introduction

Pear (*Pyrus*) is an important fruit and planted throughout the world. Red-skinned pears generally have a higher commodity value and are more favored by consumers. As a type of water-soluble flavonoid compounds, anthocyanin is responsible for the red coloration in pear peel [1]. Dietary anthocyanin has a preventive effect on cancer, cardiovascular diseases, and other chronic diseases due to its strong antioxidant capacity [2,3].

Anthocyanins are synthesized by a specific branch of the phenylpropanoid biosynthesis pathway. The precursor of anthocyanin is phenylalanine, which is catalyzed by a series of enzymes, including PAL (phenylalanine ammonia lyase), C4H (cinnamate 4-hydroxylase), 4CL (4-coumarate: CoA ligase), CHS (chalcone synthase), CHI (chalcone isomerase), F3H (flavanone 3-hydroxylase), FLS (flavonol synthase), DFR (dihydroflavonol 4-reductase), F3′H (flavanone 3′-hydroxylase), F3′5′H (flavanone 3′,5′-hydroxylase), ANS (anthocyanidin synthase), and UFGT (UDP-glucose flavonoid 3-O-glucosyltransferase) [1,2]. Although these anthocyanin biosynthesis genes are conserved, there are significant differences in the regulatory mechanisms of anthocyanin biosynthesis among plant species [1,4].

Light is considered as the key factor governing anthocyanin biosynthesis in plants [4]. The basic leucine zipper (bZIP) protein ELONGATED HYPOCOTYL 5 acts as a central regulator for the induction of anthocyanin biosynthesis [1,5]. The overexpression of HY5 promotes anthocyanin accumulation in many plant species, such as *Arabidopsis*, strawberry, and apple [6,7]. The experiments suggested that loss-of-function of HY5 could reduce the anthocyanin level in *Arabidopsis* [8], strawberry [9], and tomato [10]. As a transcription factor, HY5 accumulates in the light and directly binds to the promoters of early and late anthocyanin biosynthesis genes, such as *CHS*, *CHI*, *F3H*, *F3′H*, *DFR*, and *ANS* [8,9,10]. Furthermore, HY5 directly targets multiple downstream transcription factors that control anthocyanin biosynthesis; of these, MYB10 has been widely studied. In apple and pear, HY5 positively regulates *MYB10* expression by binding the G-box motif of the *MYB10* promoter to promote anthocyanin biosynthesis [6,11]. Since HY5 lacks a typical transactivation domain, its transcriptional activity is primarily regulated by the interaction with B-box family proteins (BBX) [12,13]. As a BBX member, a 14-nucleotide deletion mutation in the coding region of the *PpBBX24* gene is associated with the red skin of ‘Red Zaosu’ pear [14].

In addition to the light signal components, anthocyanin biosynthesis is also regulated by the MBW complex, which is composed of three protein families (MYB, bHLH, and WDR) in vivo [1,15]. In recent years, multiple MYB transcription factors have been characterized as regulators of anthocyanin biosynthesis by binding to the promoters of late anthocyanin biosynthesis genes and specifically regulating their expression [1,16,17]. In *Arabidopsis*, the overexpression of *AtMYB75* (*PAP1*) promoted anthocyanin biosynthesis, while plants downregulating *AtMYB75* and its homologs *PAP2*, *MYB113*, and *MYB114* showed obvious anthocyanin deficiencies [18]. As a homologous gene of *AtMYB75*, *MYB10* has been reported to be associated with the coloration of a variety of fruits, such as pear [19], sweet cherry [20], apple [21], and plum [22]. In apple, the expression of *MdMYB10* is highly correlated with anthocyanin levels during fruit development and the overexpression of *MdMYB10* elevates anthocyanin biosynthesis [21]. Similar to the study in apple, the expression of pear *PpMYB10* is induced by light and ectopic overexpression of *PpMYB10* in *Arabidopsis* enhances anthocyanin accumulation in immature seeds [19]. In addition to MYB10, other MYBs, such as PpMYB12 and PpMYB114, were also reported as positive regulators of anthocyanin biosynthesis [23,24].

Phytochrome-interacting factor (PIFs) is a subset of basic helix–loop–helix (bHLH) transcription factors that are involved in the light response of plants [25,26]. There are at least eight PIFs (PIF1 to PIF8) in the *Arabidopsis* genome, and most members of the PIFs family possess the conserved bHLH domain allowing them to form dimers and bind to DNA, a conserved APB motif required for the interaction with Pfr form of phyB [27,28]. Classically, members of the PIFs family act as negative regulators during photomorphogenesis. Upon light exposure, PIFs are degraded by physical interaction with the biologically active Pfr conformer, resulting in the initiation of photomorphogenic development programs, such as the inhibition of hypocotyls, the opening of apical hooks, and chloroplast development [26]. Previous studies showed that different PIF members vary in their abilities to regulate anthocyanin biosynthesis. In *Arabidopsis*, PIF3 positively regulates anthocyanin accumulation in an HY5-denpendent manner under far-red light [29], whereas PIF4 and PIF5 inhibit anthocyanin biosynthesis under red-light conditions [30]. For Rosaceae plants, PyPIF5 and MdPIF7 were reported as negative regulators in regulating anthocyanin biosynthesis in pear and apple [31,32]. However, whether there are other PIFs involved in regulating anthocyanin biosynthesis is rarely understood.

In this study, comprehensive transcriptome analysis was conducted to explore the potential regulatory mechanism of anthocyanin biosynthesis in ‘Red Zaosu’ pear. Gene co-expression analysis, transcription factor mining, and local motif enrichment analysis (CentriMo) analysis identified that light-induced PpPIF8 may play a role in anthocyanin biosynthesis. Functional studies showed that PpPIF8 localizes to the nucleus and is preferentially expressed in the tissue of higher levels of anthocyanin. The overexpression of *PpPIF8* in pear peel and pear calli promotes anthocyanin biosynthesis and upregulates the expression of anthocyanin biosynthesis genes. Yeast-one hybrid and transgenic analyses indicated that PpPIF8 binds to the *PpCHS* promoter to induce *PpCHS* expression. Taken together, our data not only provide a comprehensive view of transcription events during the coloration of pear peel, but also resolve the regulatory role of PpPIF8 in the anthocyanin biosynthesis pathway.

## 2. Results

### 2.1. Anthocyanin Accumulation in the Peel of ‘Red Zaosu’ Pear after Debagging

To investigate the response of pear fruit to light, we removed the bags at 120 days after full bloom (DAFB). The initial color of the pear peel was yellow–white. On the 5th day after debagging, the peel of D group fruits showed a slightly pink color, especially around the lenticel pit area. During the 7 to 15th day, the peel of D group fruits gradually developed an obvious red streak, while the color of B group fruits was still yellow–white (Figure 1A). Consistent with visual observation, the anthocyanin content of D group fruits increased continuously, reaching a peak at the 15th day after debagging, whereas the anthocyanin content of B control fruits remained lower during the entire period (Figure 1B). These observations suggest that light promotes the anthocyanin accumulation of ‘Red Zaosu’ pear.

### 2.2. Overview of RNA Sequencing

To dissect the regulation mechanism of anthocyanin induction by light, total RNA was extracted from B group and D group samples at 0 days (initial sample), 1, 3, 5, and 7 days after debagging. Three biological repeats were set for each timepoint. A total of 27 samples were sequenced by the Illumina platform, and 184.76 Gb of clean data were obtained, with a Q30 percentage of ≥92.97%. The clean reads were mapped to the reference genome sequence of *Pyrus bretschneideri* ‘DangshanSuli’ V1.1 [33]. The mapped rates were between 74.90% and 76.76% (Appendix A).

We conducted PCA and sample correlation analysis based on the FPKM values of all the expressed genes to evaluate the repeatability and heterogeneity of sequencing samples. As expected, the three biological replicates of each sample were clustered together, indicating good reproducibility within biological replicates. Furthermore, the samples were divided into two groups on the principal component PC1 according to bagging and debagging treatment, suggesting that light after debagging was the main factor leading to the gene expression difference between D group and B group samples. From the perspective of PC2 axis, the samples were separated by the time after debagging, which indicated that the time of debagging plays a secondary role in the heterogeneity between samples (Figure 2). Consistently, the correlation analysis of the samples exhibited very tight clustering among the three biological replicates. All B group samples exhibited a closer correlation with the initial sample (0d), while all D group samples were strikingly different (Appendix A). Taken together, these results suggest that the repeatability of sequenced samples is valid, and light after debagging, rather than the time of debagging, is the predominant contributor to gene expression differences between D group and B group samples.

### 2.3. Differential Gene Expression Analysis

To investigate the transcription differences between D and B group samples, differentially expressed genes (DEGs) were calculated at each timepoint. We used the following threshold, FDR < 0.01 and fold change ≥ 2, as the criterion. A total of unique 4658 DEGs were identified across all timepoints between D and B group samples. Overall, the numbers of upregulated genes were much higher than downregulated genes in D group samples at four different timepoints. The largest and lowest numbers of DEGs were identified at 7d and 5d, respectively. The numbers of DEGs varied from 1827 to 2591 for upregulated genes and from 290 to 514 for downregulated genes (Figure 3A and Appendix A). There were 1052 common DEGs across all timepoints, accounting for one-quarter of the total unique DEGs. The number of time-specific DEGs was 691, 363, 310, and 712 at 1, 3, 5, and 7 days after debagging (Figure 3B). These results indicate that anthocyanin accumulation after debagging is largely due to the activation of gene expression.

To evaluate RNA-Seq reliability, several anthocyanin biosynthesis genes were identified and validated by RT-qPCR. The transcript abundances of these genes were induced rapidly at 1 day after debagging but peaking at different timepoints in RNA-Seq data. However, in the B group sample, the expression of these anthocyanin biosynthesis genes remained low across all the timepoints. Among these genes, *PpCHS* (gene24418) exhibited the highest expression level after debagging (Figure 4A). Consistent with RNA-Seq data, the expression of these anthocyanin biosynthesis genes also showed a similar induction pattern by RT-qPCR in D group samples. In addition, we found that *PpCHS*, *PpFSL* (gene33758), and *PpUFGT* (gene12084) are more sensitive to light after debagging than other anthocyanin biosynthesis genes, based on their relative fold change (Figure 4B).

### 2.4. Weighted Gene Co-Expression Network Analysis (WGCNA)

#### 2.4.1. Module Construct and Module–Trait Correlation

To investigate the gene regulatory network during the coloration of pear peel, 4658 unique DEGs were subjected to weighted gene co-expression network analysis (WGCNA). A total of seven modules were identified (Figure 5A,B). Of these modules, the “Turquoise” module contained the largest gene numbers (2434 genes), followed by “Blue” module (980 genes) (Figure 5B, Appendix A).

An analysis of the module–trait relationships revealed that the “Blue” module was highly correlated with pear anthocyanin content (*r* = 0.90, *p* = 2 × 10^−10^) and *PpBBX24* expression pattern (*r* = 0.87, *p* = 5 × 10^−9^). The “Turquoise” module was positively correlated with transcript abundance of *PpHY5* (*r* = 0.77, *p* = 3 × 10^−6^), *PpMYB12* (*r* = 0.99, *p* = 2 × 10^−22^), and *PpBBX24* (*r* = 0.91, *p* = 5 × 10^−11^) (Figure 5C). The eigengene expression pattern showed that the genes in the “Blue” module were gradually induced after debagging, while the genes in the “Turquoise” module were rapidly induced in the D group samples. For both “Blue” and “Turquoise” modules, the genes in the B group samples remained unchanged (Figure 5D,E). Because genes in both “Blue” and “Turquoise” modules are positively correlated with anthocyanin accumulation and highly induced after debagging, we then combined the “Blue” and “Turquoise” gene list for further analysis afterward.

Next, we analyzed the enrichment pathways using the Kyoto Encyclopedia of Genes and Genomes (KEGG) analysis [34] on the gene list in both “Blue” and “Turquoise” modules to explore enriched biological pathways. The photosynthesis-related terms (“Photosynthesis pathway”, “Porphyrin and chlorophyll metabolism”) and the flavonoid biosynthesis-related terms (“Phenylpropanoid biosynthesis and Flavonoid biosynthesis”) were significantly enriched. In addition, the plant hormone signal transduction pathway was also substantially enriched (Figure 5F).

#### 2.4.2. Analysis of Transcription Factors

Transcription factors (TFs) play a vital role in regulating anthocyanin biosynthesis [1]. To extensively explore TFs involved in this process, the protein sequences of genes in in “Blue” and “Turquoise” modules were extracted and subjected to Plant Transcription factor & Protein Kinase Identifier and Classifier (iTAK) tools (http://itak.feilab.net/cgi-bin/itak/index.cgi, accessed on 30 June 2020) to predict TFs. A total of 263 TFs were identified and classified by transcription factor family (Table 1). The number of MYB family members was 40, followed by 35 TFs from the AP2-ERF family, 24 TFs from the WRKY family, and 18 TFs from the bHLH family. Additionally, TFs from other families, such as NAC, HB, bZIP, and B-box, were also found in the modules (Table 1).

To find the key TFs among these 263 TFs, we selected the top 55 TFs based on their KME value (KME ≥ 0.9). Among these 55 TFs, the well-known *PpMYB10* and *PpMYB12* were clustered together [22,23]. Strikingly, the *PpPIF8* (also known as *UNE10* in *Arabidopsis*) is clustered with *PpMYB10* and *PpMYB12* as well (Figure 6A), which suggests that *PpPIF8* may also participate in regulating anthocyanin biosynthesis. As expected, *PpHY5* was also strongly induced by debagging. In addition, B-box family members *PpCOL13* and *PpBBX32* were included in the modules (Figure 6A).

We next selected six TFs, including *PpHY5*, *PpMYB10*, *PpMYB12*, *PpCOL13*, *PpBBX32*, and *PpGT2*, to validate RNA-Seq results. These TFs were rapidly induced at 1d and remained relatively stable until 7d after debagging. However, the expression of these TFs did not change much in the B group samples (Figure 6B).

To further explore the association of these top 55 TFs, the protein sequences of these TFs were extracted and subjected to STRING analysis. Functional enrichment analysis of protein domains of these 55 TFs showed that the WRKY DNA-binding domain, Myb-like DNA-binding domain, and Helix–loop–helix DNA-binding domain were enriched (Figure 6C). Visualization of the STRING result showed that three clusters were identified in the network, and the first cluster (turquoise nodes) consisted of 15 members, including PpMYB10, PpMYB12, and PpHY5. The second cluster (pink nodes) contained four TFs, including PpPIF8, which connects to PpHY5 in the first cluster. The third cluster only has four genes (Figure 6C). These data indicate that the known PpMYB10 and PpMYB12 cluster with PpHY5 forms a hub network to regulate anthocyanin biosynthesis. PpPIF8, with previously unknown function in the regulation anthocyanin biosynthesis, may also participate in this process.

#### 2.4.3. Identifying Transcription Factors Regulating Gene Expression in “Blue” and “Turquoise” Modules

To discover the primary transcription factors that regulate gene expression in “Blue” and “Turquoise” modules, we extracted 2000 bp of genomic sequence located upstream of starting codon ATG of the 3158 genes in these two modules (Appendix A) and subjected them to CentriMo analysis. A total of 87 TF binding sites were found, belonging to multiple transcription factor families such as the WRKY, MYB, bZIP, and bHLH protein families (Appendix A). Surprisingly, five phytochrome-interacting factors (PIFs) were identified: PIF3, PIF4, PIF5, PIF7, and PIF8/UNE10. The potential binding sites were located within 400 bp upstream of the starting codon ATG (Figure 7 and Appendix A). This result suggests PIFs may be involved in regulating gene expression in “Blue” and “Turquoise” modules.

### 2.5. Functional Study of PpPIF8 in the Regulation of Anthocyanin Biosynthesis

In our WGCNA analysis, both transcription factor mining and CentriMo analysis [35] indicated that *PpPIF8* may play a role in regulating anthocyanin biosynthesis (Figure 6 and Figure 7). Therefore, we focused on the *PpPIF8* for functional study.

#### 2.5.1. Genome-Wide Identification of PIFs in Pear

Because the PIFs family members have not yet been systematically identified in pear, we first performed genome-wide identification of PIFs in pear. A total of eight members were identified as PIFs in the pear genome by the homology BLAST and conserved domain search. These PpPIFs were named according to their closest *Arabidopsis* homologs, and each PpPIF contained a conserved bHLH_AtPIF_like domain (Figure 8A and Table 2). We then calculated the physical and chemical properties of these PpPIFs. As shown in Table 2, PpPIF7a is the smallest member (398 amino acids), and PpPIF3a is the largest (716 amino acids). The molecular weight of the PpPIFs varies from 44 kDa (PpPIF7a) to 76.56 kDa (PpPIF3a) and their isoelectric points are between 6.09 (PpPIF3) and 9.39 (PpPIF1). The instability index of PpPIFs was larger than 53, suggesting PpPIFs may not be stable in vivo (Table 2).

To address the evolutionary relationship of PIFs in pear, apple, and *Arabidopsis*, a phylogenetic tree was constructed using full-length protein sequences. As shown in Figure 8B, the eight PpPIFs can be divided into four clades, PIF1, PIF3, PIF5, and PIF8. PpPIF1 is located in the PIF1 clade, while PpPIF3 and PpPIF3a are in the PIF3 clade, PpPIF5 and PpPIF5a are in the PIF5 clade, and PpPIF8, PpPIF7a, and PpPIF7b are in the PIF8 clade. Within each clade, PpPIFs cluster closer with apple PIFs than *Arabidopsis* PIFs. In addition, PIF2 and PIF6 members are only present in the *Arabidopsis* and apple genome, but not found in the pear genome (Figure 8B).

We then compared the expression pattern of *PpPIFs* in both B and D group samples. As shown in Figure 8C, *PpPIF1*, *PpPIF3*, and *PpPIF3a* exhibited relatively high expression levels compared with other *PpPIFs*. However, their expression levels were not observed to be regulated by light after debagging. The *PpPIF5* and *PpPIF7b* expression levels were relatively low, and they were also not regulated obviously by light. Strikingly, *PpPIF8* was rapidly and strongly induced by light, while *PpPIF7a* was only slightly induced by light at 7 d after debagging (Figure 8C). These data suggest PpPIF8 may play a unique role in regulating anthocyanin biosynthesis in pears.

#### 2.5.2. Verification of PpPIF8 Expression

To confirm the expression pattern of *PpPIF8*, RT-qPCR was performed. As shown in Figure 9A, the overall expression pattern of *PpPIF8* is similar in both RT-qPCR results and RNA-Seq data. The transcript of *PpPIF8* was rapidly and strongly induced after debagging, but remained at a low expression level in B group samples (Figure 9A).

#### 2.5.3. Tissue-Specific Expression Analysis of PpPIF8

To gain insight into the relationship between the tissue-specific expression pattern of *PpPIF8* and anthocyanin level, we extracted total RNA and anthocyanin from nine different tissues (Flower-F, Sepal-S, Young leaves-YL, Mature leaves-ML, Young fruitlet-YF, Expansion stage peel-EP, Expansion stage flesh-EF, Mature stage peel-MP, and Mature stage flesh-MF). As shown in Figure 9B, *PpPIF8* is highly expressed in young leaves (YL), which also accumulate the highest level of anthocyanin. Compared with the flesh tissue of ‘Red Zaosu’, the peel tissues of the fruit exhibit higher levels of *PpPIF8* expression and anthocyanin accumulation (Figure 9B). Furthermore, the Pearson correlation analysis showed that the expression of PpPIF8 is positively correlated with anthocyanin level (*r* = 0.7462, *p* = 0.0209), which implies that *PpPIF8* may positively regulate anthocyanin biosynthesis.

#### 2.5.4. Transcriptional Activity and Subcellular Localization Analysis of PpPIF8

To determine whether PpPIF8 has transcriptional activity, we performed transactivation activity assays in yeast. The full-length CDS of *PpPIF8* was fused in-frame with the GAL4 DNA-binding domain in the pGBKT7 vector and the construct was transformed into the Y187 yeast strain. The yeasts carrying BD-PpPIF8 or empty pGBKT7 vector could not grow on the selective medium (SD/−Trp−His), while the positive control BD-AtBZS1 grew well, indicating that PpPIF8 has no transcriptional activation activity (Figure 9C).

The correct subcellular localization of a protein is important for its function. We cloned the full-length CDS of *PpPIF8* fused with a C-terminal GFP tag under 35S promoter. The agrobacterium harboring the p35S-PpPIF8-GFP vector was co-infiltrated with nucleus marker (NSL-mCherry) strain into tobacco leaves. As Figure 9D shows, the green fluorescence of PpPIF8-GFP was detected exclusively in the nucleus and overlapped well with the red nucleus marker, whereas in the control tobacco cells transformed with an empty vector, green fluorescence was present in the nucleus and cytoplasm (Figure 9D). This result suggests that PpPIF8 is specifically localized in the nucleus.

#### 2.5.5. Overexpression of PpPIF8 Promotes Anthocyanin Accumulation in Pear

To confirm the role of PpPIF8 in anthocyanin biosynthesis in pear, we transiently overexpressed *PpPIF8* in the peel of ‘Red Zaosu’ pear. After 5 days of light treatment, we found that the overexpression of *PpPIF8* induced anthocyanin biosynthesis surrounding the injection site, while no red color was observed in the injection area of the empty vector (Figure 10A). Furthermore, we induced pear calli from the flesh cells of the young fruitlet of red-skinned *Pyrus communis* ‘Xiuzhenxiang’ and overexpressed the *PpPIF8* (*PpPIF8_OE*) in pear calli. The coloration phenotype was observed after confirming the transgene by immunoblotting (Figure 10C). As expected, the color of *PpPIF8_OE* calli turned red after 5 days of light treatment, while the color of non-transgenic control (CK) calli stayed pale yellow (Figure 10B). Consistent with the phenotype observation, the anthocyanin content of *PpPIF8_OE* calli was 10-fold higher than that of the CK calli (Figure 10D). Together, these data suggest that *PpPIF8* positively regulates anthocyanin biosynthesis.

#### 2.5.6. RNA-Seq Analysis of *PpPIF8_OE* Transgenic Pear Calli

To further explore the mechanism of *PpPIF8* in regulating anthocyanin biosynthesis, the light-grown *PpPIF8_OE* and the non-transgenic control pear calli were used for deep RNA-Seq analysis. Compared with the control sample, 923 genes were significantly upregulated and 707 genes were downregulated in *PpPIF8_OE* pear calli (Figure 11A). The anthocyanin biosynthesis genes *PpPAL* (gene3545), *PpC4H* (gene22826), *Pp4CL* (gene20027), *PpCHS* (gene24418), *PpCHI* (gene40086), *PpF3H* (gene13883), *PpF3′H* (gene3191), *PpFLS* (gene33758), *PpANS* (gene13320), and *PpUFGT* (gene12084) were all upregulated in *PpPIF8_OE* pear calli, of which *PpCHS* had the smallest FDR value (Figure 11A). Consistent with RNA-Seq data, the RT-qPCR results also showed a similar expression trend (Figure 11B). Besides anthocyanin biosynthesis genes, both RNA-Seq and RT-qPCR results show that the transcription factors *PpMYB10* and *PpMYB12*, but not *PpHY5*, were also upregulated, whereas *PpCOL13* and *PpGT2* were slightly downregulated in *PpPIF8_OE* pear calli (Figure 11C). These results suggest that PpPIF8 may function upstream of PpMYB10 and PpMYB12.

Furthermore, the KEGG enrichment analysis was conducted for DEGs. For the PpPIF8 upregulated genes, the terms significantly enriched were: Flavonoid biosynthesis, Secondary metabolites biosynthesis, Plant hormone signal transduction, Photosynthesis proteins. However, for the PpPIF8 downregulated genes, the terms significantly enriched were: MAPK pathway, Transcription factor, Signal transduction, Plant–pathogen interaction (Figure 11D). Together, these data suggest that PpPIF8 promotes anthocyanin biosynthesis mainly by activating gene expression, such as *PpCHS*, *PpMYB10*, and *PpMYB12*.

CHS is a key enzyme in the anthocyanin biosynthesis pathway. However, in the RNA-Seq result of *PpPIF8_OE* pear calli, *PpCHS* showed the smallest FDR value among other anthocyanin biosynthesis genes (Figure 11A). Thus, we speculated whether PpPIF8 could directly regulate *PpCHS*. To test this hypothesis, we cloned the promoter sequence of *PpCHS* (*proPpCHS*, 1600 bp upstream of the starting codon ATG) from ‘Red Zaosu’ and predicted the potential cis-acting regulatory elements using the PlantCARE program. Multiple light-responsive elements were detected, such as G-box, Box 4, TCT motif, TCCC motif, Sp1 motif, and chs-CMA1a motif (Figure 11E). It is reported that in *Arabidopsis*, PIF3 binds to the G-box of CHS. Therefore, we speculated that G-box may be also important for PpPIF8 binding to *PpCHS* promoter of which G-box is considered as PIF binding motif in previous literature [29]. We then performed a yeast-one hybrid assay to test if PpPIF8 can directly bind to the *PpCHS* promoter. As Figure 11F shows, the yeast cells co-transformed with AD-PpHY5 and *proPpCHS*-pHIS2.1 vectors were set as positive control and could grow on both SD/−Leu−Trp and SD/−Leu−Trp−His plus 5 mM 3-AT mediums. As with the positive control, yeast cells co-transformed with AD-PpPIF8 and *proPpCHS*-pHIS2.1 vectors could also grow on both selective mediums. However, the negative control yeast cells co-transformed with empty AD and *proPpCHS*-pHIS2.1 vectors could only grow on SD/−Leu−Trp medium (Figure 11F). This result demonstrates that PpPIF8 can directly bind to the *PpCHS* promoter.

## 3. Discussion

### 3.1. Light Is a Key Factor for Anthocyanin Biosynthesis in Pear Peel

Bagging is an effective approach to improve fruit quality and appearance. However, bagging seriously affects the coloration of fruit peel due to blocking light penetration. Therefore, light is considered as a key environmental factor that regulates anthocyanin biosynthesis [36]. Generally, anthocyanin content and the expression of anthocyanin biosynthesis genes is lower in bagged fruits compared with debagged fruits. In this study, we found that the peel of debagged ‘Red Zaosu’ pear turns red gradually under light, while the peel of bagged fruits stays pale white (Figure 1), indicating that light is indispensable for “Red Zaosu” coloration. These data are consistent with previous observations of pear and apple [36,37].

### 3.2. Comprehensive RNA-Sequencing and Gene Co-Expression Network Analysis Identify Potential TFs Involved in Anthocyanin Biosynthesis

RNA-Sequencing (RNA-Seq) is a powerful tool for dissecting the transcription events during anthocyanin biosynthesis [36,38]. Here, we performed time-series RNA-Seq using ‘Red Zaosu’ pear, and 4658 unique DEGs were identified after debagging (Figure 3), suggesting extensive transcriptional reprogramming occurs after exposure to light. Among these DEGs, multiple anthocyanin biosynthesis genes were induced by light after debagging (Figure 4), which is consistent with previous research [36].

WGCNA is a powerful tool to construct gene co-expression network and calculate module-trait relationship [39]. In our study, seven modules were identified and the “Blue” and “Turquoise” modules were highly correlated with anthocyanin biosynthesis (Figure 5A–C). KEGG enrichment analysis revealed that genes in these two modules are related to light-regulated pathways, such as chlorophyll metabolism, photosynthesis, and flavonoid biosynthesis (Figure 5F), suggesting light after debagging is the main cause of these physiological changes.

Transcription factors play an important role in regulating anthocyanin biosynthesis, and many TFs have been identified during coloration of pear peel [1,36]. In our study, 263 TFs were identified in the “Blue” and “Turquoise” modules (Table 1). Consistent with previous studies, some key TFs, such as PpHY5, PpMYB10, and PpMYB12, were also included (Figure 6A). Moreover, there are also many TFs have not yet been characterized (Figure 6A); these TFs can direct subsequent studies. Further cluster study found that PpPIF8 clustering was highly correlated with PpMYB10 and PpMYB12 (Figure 6A). STRING analysis also indicated PpPIF8 is associated with PpHY5 (Figure 6C). CentriMo analysis showed that PIFs are enriched in the promoters of genes in the “Blue” and “Turquoise” modules (Figure 7 and Appendix A). These results suggest that PpPIF8 may play a role in regulating anthocyanin biosynthesis.

### 3.3. Identification of PIFs in Pear Genome

PIFs act downstream of phytochrome to regulate a range of photomorphogenic development processes, such as hypocotyl elongation, cotyledon opening, and anthocyanin biosynthesis [26]. We first identified PIFs in the pear genome. Like *Arabidopsis* and apple, there are eight PpPIFs in pear genome, and each of them contains a conserved bHLH-AtPIF-like domain (Figure 8A). Surprisingly, in the comparison of expression patterns, only *PpPIF8* was strongly and quickly induced by light after debagging (Figure 8C and Figure 9A), which is similar to *AtPIF8* in *Arabidopsis* and *VvPIF7* in grape skin [40,41]. These data suggest that PpPIF8 transcripts may play a unique role among these PpPIFs in regulating anthocyanin biosynthesis in pear.

### 3.4. Tissue-Specific Expression, Subcellular Localization, and Transcriptional Activity of PpPIF8

Tissue-specific expression analysis showed that *PpPIF8* is preferentially expressed in the tissue with higher anthocyanin content, with a correlation coefficient of 0.7462. Strikingly, as for the fruits, the expression level of *PpPIF8* in pear peel is over 50-fold greater than pear flesh (Figure 9B). Similarly, *VvPIF7* is also highly expressed in the grape peel tissue, but not flesh, at the pre-*verasion* and post-*verasion* stages [41]. Collectively, these data suggest that tissue-specific expression of PpPIF8 may participate in regulating anthocyanin biosynthesis in the peel of pear fruit.

As transcription factors, nuclear localization and transcription activity are essential for their function. A co-localization assay showed that PpPIF8 was expressed specifically in the nucleus (Figure 9D). This is consistent with MdPIF8 in apple [42]. In addition, transactivation activity assays in yeast cells showed that PpPIF8 does not have self-activation activity, suggesting that other transcriptional regulators may be required for its transcription activity in vivo (Figure 9C). In support of this speculation, MdPIF7 also lacks self-activation activity, but interacts with MdBBX23, which directly regulates MdHY5 expression [32]. Therefore, the identification of the PpPIF8-interacting partners in the future may help elucidate how PpPIF8 regulates gene expression in vivo.

### 3.5. PpPIF8 Positively Regulates Anthocyanin Biosynthesis

To investigate the function of PpPIF8 in the anthocyanin biosynthesis pathway, we overexpressed PpPIF8 by transient infiltration in pear peel and stable transformation of pear calli. Both results showed that PpPIF8 promotes anthocyanin biosynthesis in light (Figure 10). This is consistent with the role of *Arabidopsis* PIF3, but in contrast to PIF4 and PIF5 [29,30]. In *Arabidopsis*, the overexpression of PIF3 induces anthocyanin accumulation, while *pif3* mutants showed decreased anthocyanin levels in far-red light [29]. Unlike PIF3, *Arabidopsis* PIF4 and PIF5 negatively regulate anthocyanin accumulation under red light [30]. During our research, two PIFs were published as negative regulators in regulating anthocyanin biosynthesis in apple and pear [31,32]. In apple, the overexpression of MdPIF7 decreased anthocyanin accumulation in transgenic apple calli and MdPIF7 antisense suppressing apple calli increased anthocyanin [32]. In red Chinese sand pear ‘Yunhongyihao’, *PyPIF5* is downregulated by light after debagging and negatively regulates anthocyanin accumulation through the PyPIF5–*PymiR156a*–PySPL9–PyMYB114/MYB10 cascade [31]. However, in our data, the expression of both *PpPIF5* and *PpPIF5a* is not obviously different between bagged and debagged samples (Figure 8C). We speculate that this discrepancy may arise for at least two reasons. The first is that different pear varieties were used. The strip-colored ‘Red Zaosu’ pear, which is a specific PpBBX24 mutation pear [14], is used in our study, whereas “Yunhongyihao” is an evenly colored pear. The second possible reason is due to the difference in temperature and light intensity and quality between south and north China. In summary, as for anthocyanin biosynthesis, different PIFs function as either positive or negative regulators under different light spectra.

### 3.6. Molecular Mechanisms of PpPIF8 in Regulation of Anthocyanin Biosynthesis

To investigate how PpPIF8 promotes anthocyanin biosynthesis, DEGs were identified by RNA-Seq in *PpPIF8_OE* transgenic pear calli and the control calli. In accord with the function of PpPIF8 in promoting anthocyanin biosynthesis, multiple anthocyanin biosynthesis genes, *PpMYB10*, and *PpMYB12* are upregulated in *PpPIF8_OE* transgenic pear calli (Figure 11A). KEGG enrichment analysis showed that some light-regulated terms are enriched in PpPIF8-upregulated genes, such as ‘Flavonoid biosynthesis’ and ‘Photosynthesis protein’ (Figure 11D). These data suggest PpPIF8 has a positive role in light-regulated processes. Similar to the PIF3 in *Arabidopsis* [29], PpPIF8 can also directly binds the promoter sequence of *PpCHS* (Figure 11F). Therefore, PpPIF8 promotes anthocyanin biosynthesis probably through direct activation of the expression of anthocyanin biosynthesis genes. Previous research showed that the red color of ‘Red Zaosu’ was highly associated with PpBBX24 mutation, and that this mutation was possibly specific to ‘Red Zaosu’ [14]. Furthermore, in *Arabidopsis*, the AtBBX24 interacts with AtHY5 and negatively regulates its transcriptional activity [43], and the promotion effect of AtPIF3 on anthocyanin biosynthesis is also dependent on AtHY5 [29]. Therefore, we speculated that PpBBX24 mutation in ‘Red Zaosu’ might derepress PpHY5 activity, and the promotion effect of PpPIF8 on anthocyanin biosynthesis might depend on this elevated PpHY5 activity in the ‘Red Zaosu’ background.

In summary, our data not only provide a comprehensive view of transcription events during the coloration of pear peel, but also reveal a simple molecular mechanism of PpPIF8 in the regulation of anthocyanin biosynthesis in pear. Upon exposure to light, *PpPIF8* is upregulated and then directly binds the promoter sequence of *PpCHS* to induce its expression. In addition, PpPIF8 may regulate the expression of *PpMYB10* and *PpMYB12* in a currently unknown way. Further efforts should be paid to identify PpPIF8’s partners in vivo and clarify how *PpMYB10* and *PpMYB12* are regulated by PpPIF8.

## 4. Materials and Methods

### 4.1. Plant Materials and Treatments

The fruits of ‘Red Zaosu’ (*Pyrus pyrifolia* × *Pyrus communis*) were harvested from Zhao County, Shijiazhuang City, Hebei Province. In total, 300 fruits were covered with lightproof double-layered paper bags at 40 days after full blossom (DAFB). Half of the fruits were debagged at 120 DAFB (D group), and the remaining bagged fruits were used as the control (B group). One hundred and fifty fruits of both D and B groups were randomly divided into three biological replicates and sampled at 1, 3, 5, 7, and 15 days after debagging. The fruits just before debagging were regarded as the 0-day sample. At each sampling time point, the fruits were photographed, and fruit peels were scraped, immediately frozen in liquid nitrogen, and stored at −80 °C until use.

### 4.2. Induction and Transformation of Pear Calli

The pear calli were induced from the flesh cells of the young fruitlet of red-skinned *Pyrus communis* ‘Xiuzhenxiang’, as reported previously [44] and subcultured on Murashige and Skoog (MS) solid medium (Coolaber, Beijing, China) supplemented with sucrose, 2,4-dichlorophenoxyacetic acid, and 6-benzylaminopurine at 22 °C under dark conditions.

The coding sequence of *PpPIF8* without the stop codon was amplified from cDNA of debagged fruit sample, cloned into pENTR/SD/D-TOPO vector (named PpPIF8-pTOPO afterward) (Invitrogen, Waltham, MA, USA), and then recombined into pGWB5, which contains a C-terminal GFP tag destination vector by LR CLONASE Enzyme Mix (Invitrogen, Waltham, MA, USA). The resulting vector (*PpPIF8_OE*) was confirmed by sequencing and then introduced into *Agrobacteria tumefaciens* strain EHA105 (Weidi Biotechnology, Shanghai, China) using the freeze–thaw method. For the transformation of pear calli, agrobacterium carrying the *PpPIF8_OE* vector was cultured until an OD_600_ of 0.6 in LB liquid medium supplied with 50 µg/mL kanamycin and 50 µg/mL rifampicin. The EHA105 cells were collected by centrifugation at 5000× *g* for 15 min and then resuspended in transformation buffer (MS liquid medium, 10 mmol/L MES, 200 μmol/L Acetosyringone, pH 5.6) to a final OD_600_ of 0.5. The pear calli were incubated with *A. tumefaciens* carrying *PpPIF8_OE* vector for 15 min. After co-culture on MS solid medium for 2 days, the calli were then screened on MS solid medium containing 30 mg/L hygromycin under continuous dark conditions at 22 °C. For the light treatment, the freshly cultured *PpPIF8_OE* and non-transgenic pear calli were exposed to light (light intensity: 100 mmol m^−2^ s^−1^; photoperiod: 16 h light/8 h dark) for 5 days and then used for analysis.

The overexpression of *PpPIF8* in transgenic pear calli was confirmed by immunoblotting. In brief, total protein was extracted with 2 × SDS buffer (100 mM Tris–HCl, pH = 6.8; 20% glycerol; 4% SDS; 2% β-mercaptoethanol; 0.01% bromophenol blue). The samples were boiled at 95 °C for 15 min and centrifuged at 13,000× *g* for 10 min. The denatured samples were separated by 10% SDS-PAGE and transferred onto a PVDF membrane (Millipore, Burlington, MA, USA). The membrane was blocked with 5% non-fat milk followed by antibody incubation. Chemiluminescence signals were visualized using SuperSignal West Dura Extended Duration Substrate (Thermo Scientific, Waltham, MA, USA) and X-ray film. The GFP monoclonal antibody (HT801-01) and HRP-conjugated Goat anti-mouse secondary antibody (HS201-01) were purchased from TransGene (Beijing, China).

### 4.3. Anthocyanin Extraction and Measurement

The anthocyanin content was measured according to a previous report [45]. Briefly, the fruit peel, calli, and different tissues of pear were ground into fine powder in liquid nitrogen. For each sample, 0.2 g tissue powder was extracted in the dark with 5 mL methanol–HCl (99:1, *v*/*v*) at 4 °C overnight. The extract was then centrifuged at 13,000× *g* for 15 min, and the absorbances of the resulting supernatant was measured by a UV–visible spectrophotometer UV2600 (Shimadzu, Kyoto, Japan) at wavelengths of 530 and 657 nm, respectively. The anthocyanin content was calculated as follows: anthocyanin content = (A_530_ − 0.25 × A_657_) × *v*/*w*, where *v* = volume of the extract (mL) and *w* = weight of tissue powder (g).

### 4.4. RNA Extraction and RT-qPCR

Total RNA was extracted using RNAprep Pure Plant Plus Kit (for polysaccharides and polyphenolic-rich samples) (TIANGEN, Beijing, China) according to the manufacturer’s instructions. RNA purity and concentration were determined using a NanoDrop spectrophotometer (Thermo Scientific, Waltham, MA, USA). RNA integrity was evaluated using agarose gel electrophoresis. Reverse transcription was performed using PrimeScript^TM^ RT Reagent Kit with gDNA Eraser (Takara, Kusatsu, Japan) and real-time fluorescence quantitative PCR (RT-qPCR) was conducted using TB Green Premix Ex Taq^TM^ II (Tli RNaseH Plus) kit (Takara, Kusatsu, Japan) on a 7500 Real-Time PCR System (Applied Biosystems, Waltham, MA, USA). *PpACTIN7* was used as the internal reference. Three biological repeats were performed for each sample and the relative expression levels of genes were calculated by the 2^−ΔΔCt^ method [46]. Primers used for RT-qPCR are shown in Appendix A.

### 4.5. RNA-Seq Analysis and WGCNA

Total RNA was extracted from both D and B group samples and each time point set with three biological repeats. RNA quality was evaluated by agarose gel electrophoresis and Agilent Bioanalyzer 2100 system (Agilent Technologies, Santa Clara, CA, USA). The libraries were generated using the NEBNext Ultra^TM^ RNA Library Prep Kit for Illumina (NEB, Ipswich, MA, USA) and then sequenced with the Illumina NovaSeq 6000 (Illumina, San Diego, CA, USA) by Biomarker Technologies Corporation (Beijing, China). The clean reads were mapped to the reference genome of *Pyrus bretschneideri* ‘DangshanSuli’ V1.1 by HISAT2 software [33,47]. The gene expression levels were calculated based on the number of fragments per kilobase of transcript per million reads mapped (FPKM). Differential expression analysis was performed using the R package DESeq2 [48]. PCA and sample correlation were analyzed in R software (version 3.6) and visualized using ggplot2 package in R and TBtools, respectively [49]. WGCNA was performed using the WGCNA package in R [39]. KEGG enrichment analysis was completed by TBtools and plotted using ggplot2 package in R.

For the STRING analysis, the protein sequences were uploaded on the server (https://cn.string-db.org, accessed on 23 August 2021). *Pyrus*
*× bretschneideri* was set for the organism option and the default setting for other parameters. The network was visualized using Cytoscope software (version 3.9.0) [50].

For local motif enrichment analysis (CentriMo), promoter sequences (2000 bp upstream of the starting codon ATG) were extracted by TBtools and then analyzed using the online tool CentriMo (version 5.2.0. https://CentriMo-suite.org/CentriMo/tools/centrimo, accessed on 26 November 2021). The JASPAR CORE (2018) plants motif database was selected to identify enriched known motifs in these promoter sequences [35].

### 4.6. Identification of the PpPIF Genes and Phylogenetic Tree Construction

The protein sequences of eight *Arabidopsis* PIFs were retrieved from The Arabidopsis Information Resource (TAIR 10) (https://www.arabidopsis.org/, accessed on 10 December 2021) and used as query sequences to search homologs *Pyrus bretschneideri* ‘DangshanSuli’ V1.1 genome database by BLAST using TBtools software [33,49]. The obtained sequences were further searched in the Conserved Domains Database (https://www.ncbi.nlm.nih.gov/CDD, accessed on 12 December 2021) to confirm whether the candidate PpPIFs contain the typical bHLH_AtPIF_like domain (cd11445). The molecular weights, isoelectric points, and instability index of PpPIFs were calculated using ExPASy ProParam tool (https://web.expasy.org/protparam/, accessed on 15 December 2021).

For constructing the phylogenetic tree, the PIFs protein sequences from *Arabidopsis*, apple, and pear were aligned with Multiple Protein Sequence Alignment (MUSCLE) using the default settings in MEGA version 11 software [51], and a phylogenetic tree was constructed by applying the maximum likelihood method with 1000 bootstrap repeats using MEGA version 11 software [51].

### 4.7. Subcellular Localization Analysis

The full-length CDS of *PpPIF8* without stop codon was cloned into pCambia1300-GFP vector under 35S promoter using *pEASY*-Basic Seamless Cloning and Assembly Kit (TransGene, Beijing, China). The empty vector was used as a control. The resulting p35S-PpPIF8-GFP vector was transformed into *Agrobacterium tumefaciens* (GV3101) and then co-infiltrated with nuclear marker (NSL-mCherry) strain into *Nicotiana benthamiana* leaves. After 48 h of infiltration, the fluorescence signal was detected using a laser confocal microscopy (Leica TCS SP8, Wetzlar, Germany).

### 4.8. Transient Overexpression Assay in ‘Red Zaosu’ Fruit

Transient overexpression assay in ‘Red Zaosu’ fruit was conducted as previously described, with minor modifications [52]. The bagged ‘Red Zaosu’ pear fruit, which was harvested at 132 DAFB, was used for transient overexpression assay. The *A. tumefaciens strains* carrying p35S-PpPIF8-GFP and empty vectors separately were cultured in LB medium overnight. The cells are harvested by centrifugation and resuspended in infiltration buffer (10 mM MgCl_2_, 10 mM MES, and 150 mM Acetosyringone) to a final OD_600_ of 0.8. The resuspended cell was injected into ‘Red Zaosu’ pear fruit. The fruits were then incubated in the dark for two days and transferred to continuous white light condition for observation.

### 4.9. Transactivation Activity Assay

Transactivation activity assay was performed as previously described with minor modifications [52]. The BD-PpPIF8 vector was constructed by recombining MluI-digested PpPIF8-pTOPO vector into a gateway-compatible pGBKT7 (BD) vector using LR CLONASE Enzyme Mix (Invitrogen, Waltham, MA, USA). The BD-PpPIF8, together with the negative controls (empty BD) and positive control (BD-AtBZS1), were individually transformed into yeast strain Y187 using the Super Yeast Transformation Kit ΙΙ (Coolaber, Beijing, China). The yeast transformants were sequentially screened on SD/−Trp and SD/−Trp/−His plates. The transactivation activity was determined by yeast growth on SD/−Trp/−His plates.

### 4.10. Yeast One-Hybrid Assay

The promoter sequence of *PpCHS* (gene24418) was cloned from ‘Red Zaosu’ pear using Universal GenomeWalker 2.0 kit (Clontech, Mountain View, CA, USA) and ligated into pHIS2.1 vector by *pEASY*-Basic Seamless Cloning and Assembly Kit (TransGene, Beijing, China). The resulting plasmids *proPpCHS*-pHIS and AD-PpPIF8 were co-transformed into Y187 strain using the Super Yeast Transformation Kit ΙΙ (Coolaber, Beijing, China). The transformants were successively screened on SD/−Leu−Trp plates and tested on SD/−Leu−Trp−His plates plus 5 mM 3-amino-1,2,4-triazole(3AT) at 30 °C for 3 days. Co-transformants containing pGADT7 and *proPpCHS*-pHIS or AD-PpHY5 and *proPpCHS*-pHIS were used as negative and positive control, respectively.

### 4.11. Statistical Analysis

Statistical analysis was performed with GraphPad Prism software (version 9). Significant differences (* *p* < 0.05, ** *p* < 0.01, and *** *p* < 0.001) were determined with Student’s *t*-test.

## Figures and Tables

**Figure 1 ijms-23-04798-f001:**
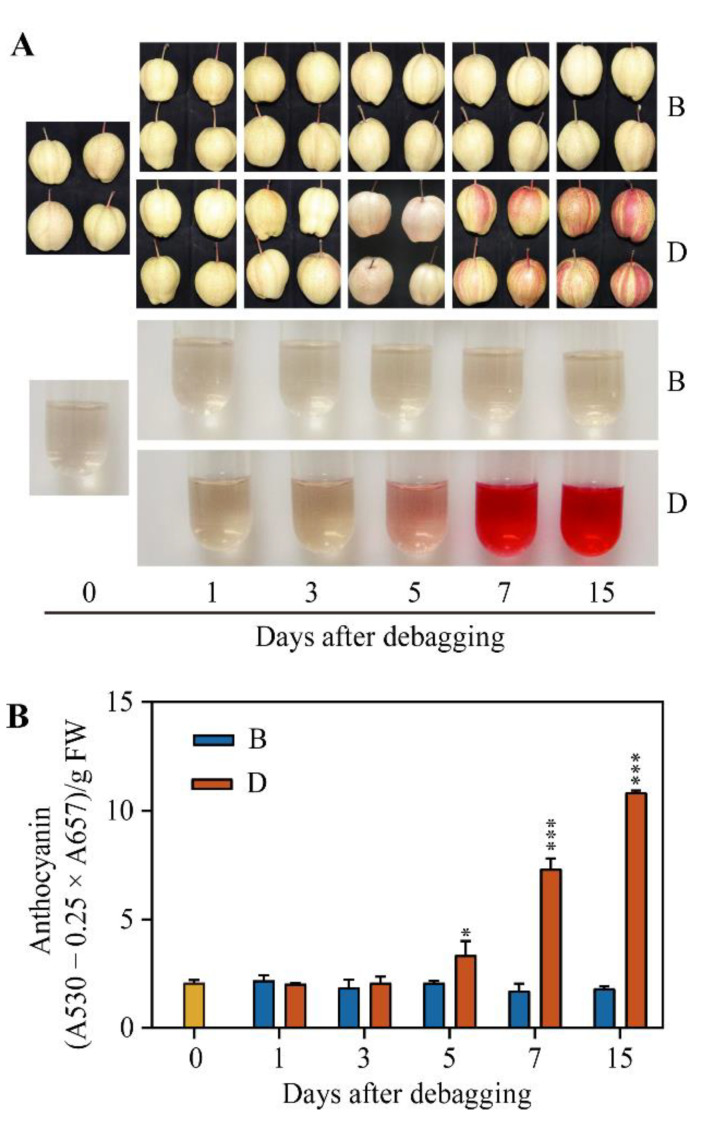
Debagging promotes anthocyanin accumulation in the peel of ‘Red Zaosu’ pear. (**A**) Color changes in ‘Red Zaosu’ pear. ‘0’ was the initial sample before debagging. ‘B’ represents bagging samples, while ‘D’ indicates debagging samples. (**B**) Changes in the anthocyanin content after debagging. Error bars denote the standard deviations. Asterisks above the bars indicate significant differences (*** *p* < 0.001, * *p* < 0.05) obtained by two-tailed Student’s *t*-test within each timepoint.

**Figure 2 ijms-23-04798-f002:**
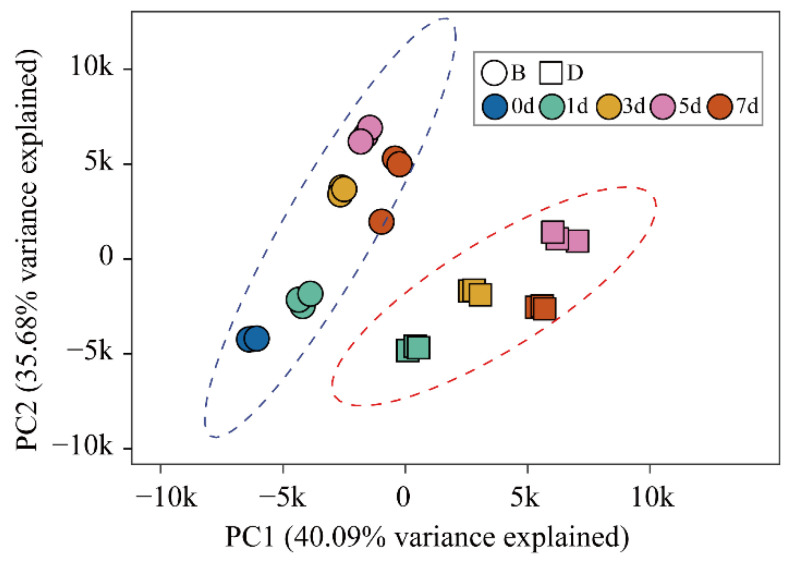
Principal component analysis (PCA) plots showing repeatability and heterogeneity of sequencing samples.

**Figure 3 ijms-23-04798-f003:**
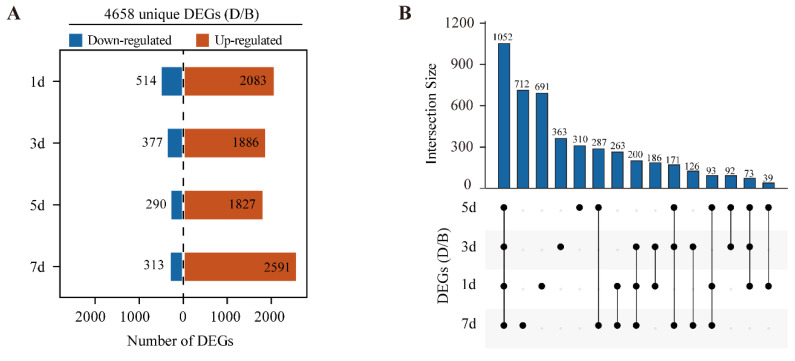
Differential gene expression analysis. (**A**) Counts of DEGs. (**B**) Upset plot showing the intersection of DEGs across all timepoints.

**Figure 4 ijms-23-04798-f004:**
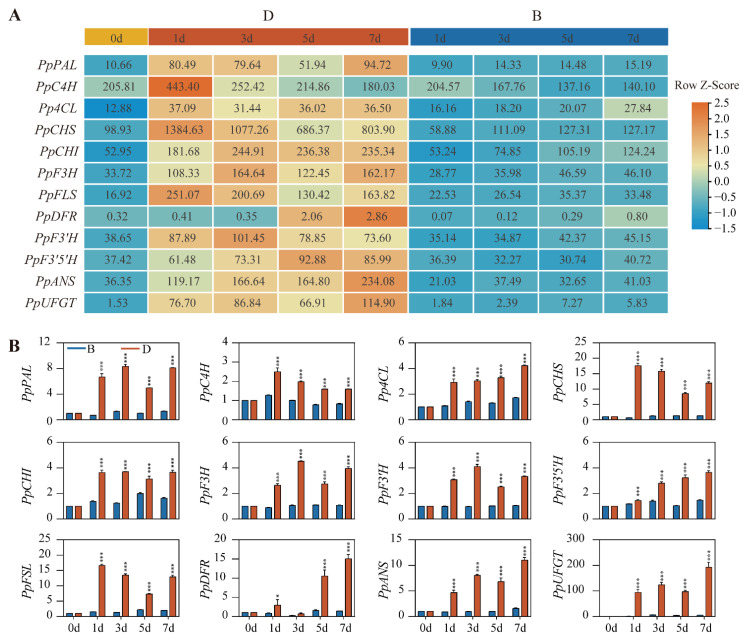
Expression of anthocyanin biosynthesis genes. (**A**) Heatmap illustrating the expression of anthocyanin biosynthesis genes. The numbers within the grids are original FPKM values, whereas row-scaled FPKM values are used for plotting. (**B**) Validation of gene expression by RT-qPCR. Error bars denote the standard deviations. Asterisks above the bars indicate significant differences (*** *p* < 0.001, * *p* < 0.05) obtained by two-tailed Student’s *t*-test within each timepoint.

**Figure 5 ijms-23-04798-f005:**
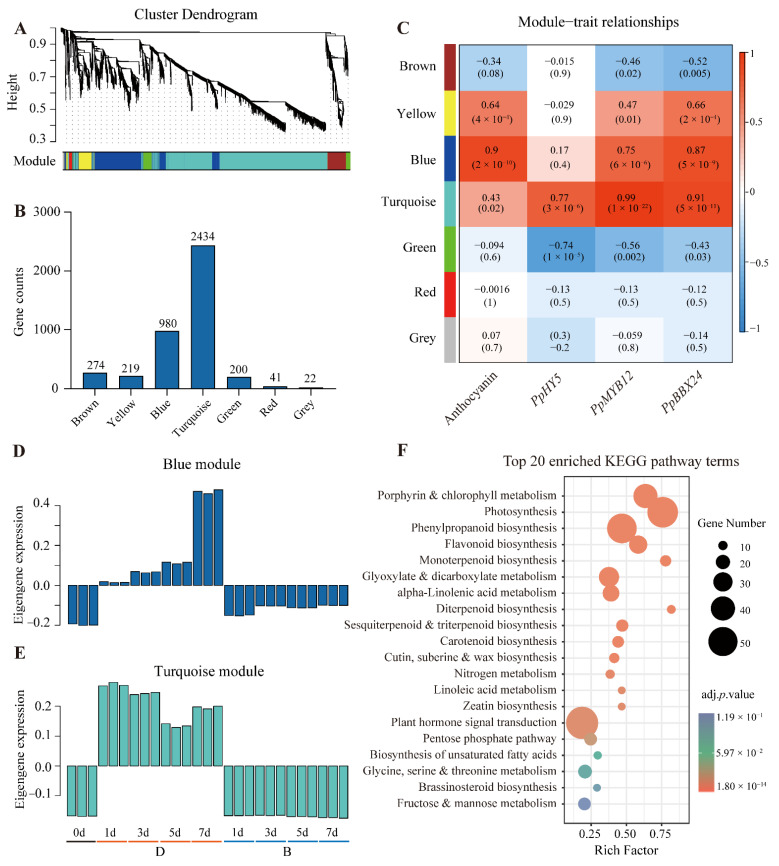
WGCNA and KEGG enrichment analysis. (**A**) Module construction of WGCNA. (**B**) Genes counts in each module. (**C**) Heatmap showing module–trait relationships. The numbers in each grid are Pearson correlation coefficient and significant *p* value in brackets. (**D**) Eigengene expression in “Blue” module. (**E**) Eigengene expression in “Turquoise” module. (**F**) Scatter plot showing top 20 enriched KEGG pathways in “Blue” and “Turquoise” modules.

**Figure 6 ijms-23-04798-f006:**
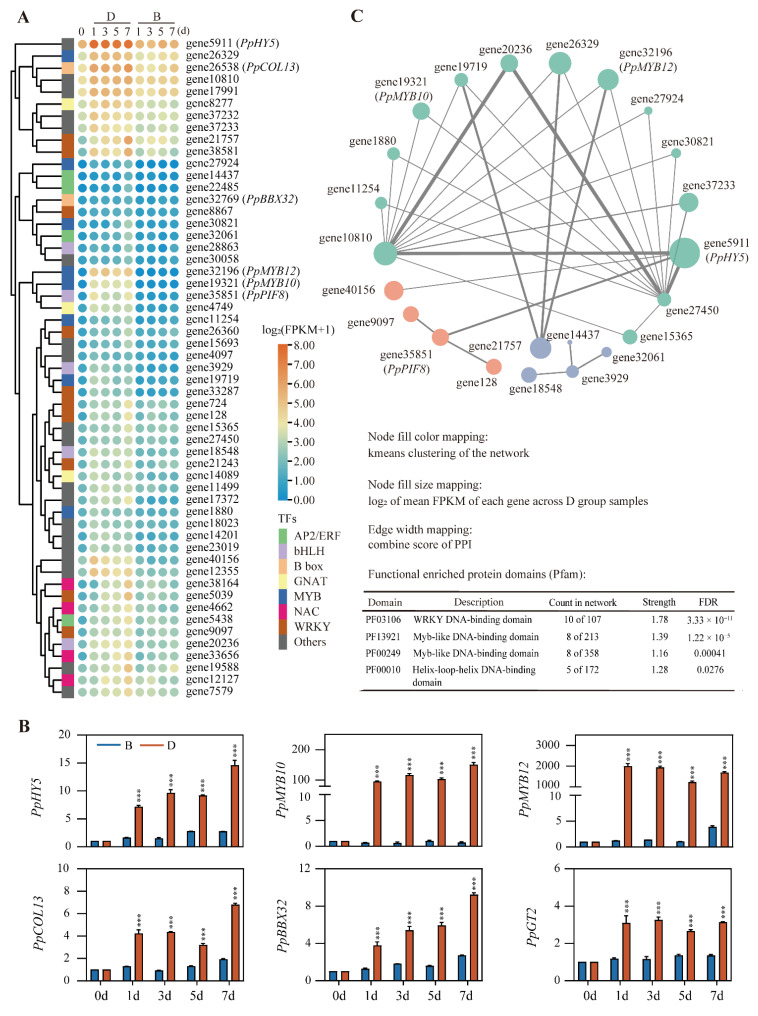
Analysis of transcription factors. (**A**) Cluster analysis of 55 transcription factors expression in “Blue” and “Turquoise” modules. (**B**) Validation of transcription factor expression by RT-qPCR. Error bars denote the standard deviation. Asterisks above the bars indicate significant differences (*** *p* < 0.001) obtained by two-tailed Student’s *t*-test within each timepoint. (**C**) STRING-DB analysis of the transcription factors.

**Figure 7 ijms-23-04798-f007:**
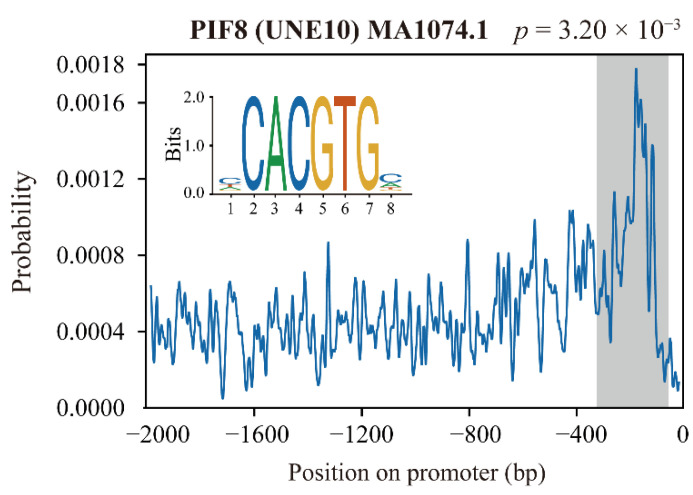
The distribution preference of the PIF8 binding site in the promoter region of genes in the “Blue” and “Turquoise” modules.

**Figure 8 ijms-23-04798-f008:**
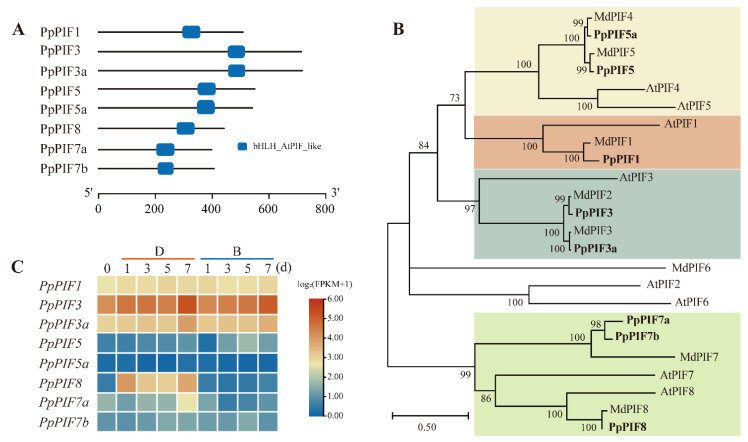
Genome-wide identification of PpPIFs. (**A**) PpPIFs contains a conserved bHLH_AtPIF_like domain. (**B**) Phylogenetic tree analysis of PIFs from pear, apple, and *Arabidopsis thaliana*. (**C**) The expression pattern of *PpPIF1*, *PpPIF3*, *PpPIF5*, and *PpPIF8*. ‘B’ represents bagged samples, while ‘D’ indicates debagged samples.

**Figure 9 ijms-23-04798-f009:**
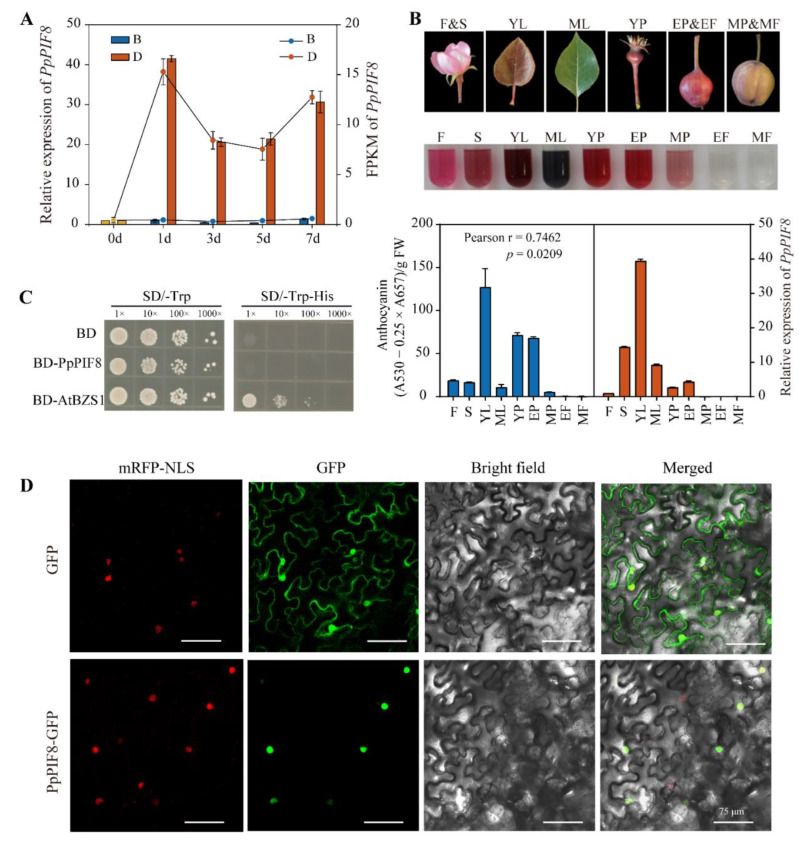
Expression, transcription activation, and subcellular localization of PpPIF8. (**A**) Verification of *PpPIF8* expression in RT-qPCR results and RNA-Seq data. (**B**) Expression patterns of *PpPIF8* in different tissues. The abbreviation in the graph are as follows: Flower-F, Sepal-S, Young leaf-YL, Mature leaf-ML, Young fruitlet-YF, Expansion stage peel-EP, Expansion stage flesh-EF, Mature stage peel-MP, and Mature stage flesh-MF. (**C**) Transcriptional activation of PpPIF8 using a yeast assay. The GAL4 DNA-binding domain (**B**,**D**) alone was used as the negative control. The BD-AtBZS1 was used as the positive control. SD/−Trp, synthetic dextrose medium lacking Trp; SD/−Trp-His, synthetic dextrose medium lacking both Trp and His. (**D**) Subcellular localization of PpPIF8 in *Nicotiana benthamiana* leaf cells. Scale bar: 75 μm.

**Figure 10 ijms-23-04798-f010:**
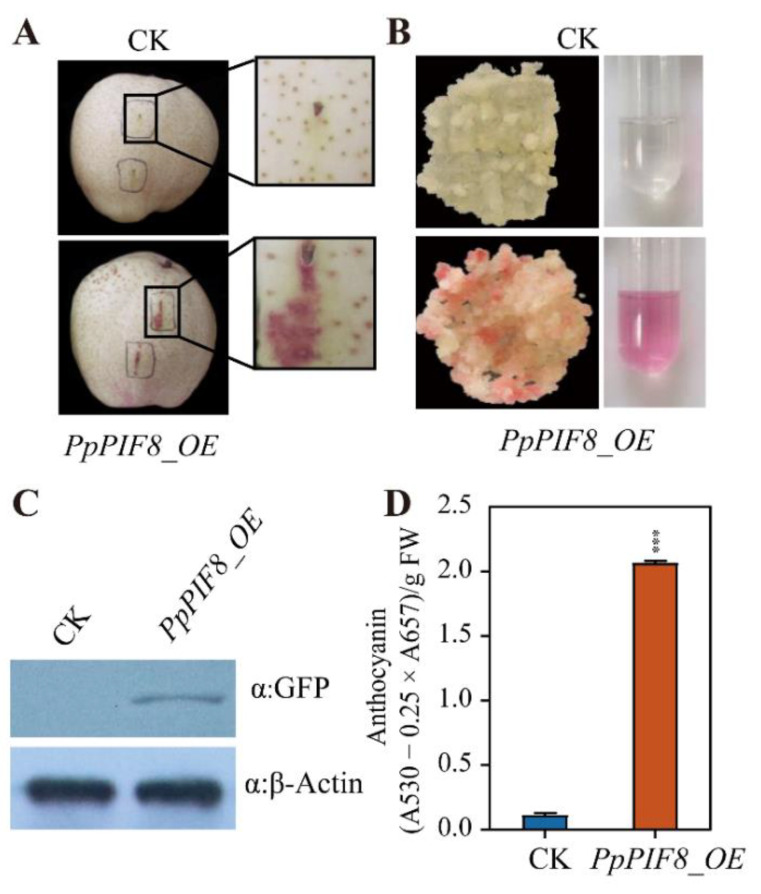
Overexpression of *PpPIF8* promotes anthocyanin accumulation. (**A**) Transiently overexpression of *PpPIF8* in the peel of ‘Red Zaosu’ pear; empty vector was used as the negative control. (**B**) Overexpression of *PpPIF8* promotes anthocyanin accumulation in pear calli. (**C**) Conformation of PpPIF8 overexpression by immunoblotting; β-actin was used as an internal reference. (**D**) Determination of anthocyanin contents in (**B**). Error bars denote the standard deviations. Asterisks above the bars indicate significant differences (*** *p* < 0.001) obtained by two-tailed Student’s *t*-test.

**Figure 11 ijms-23-04798-f011:**
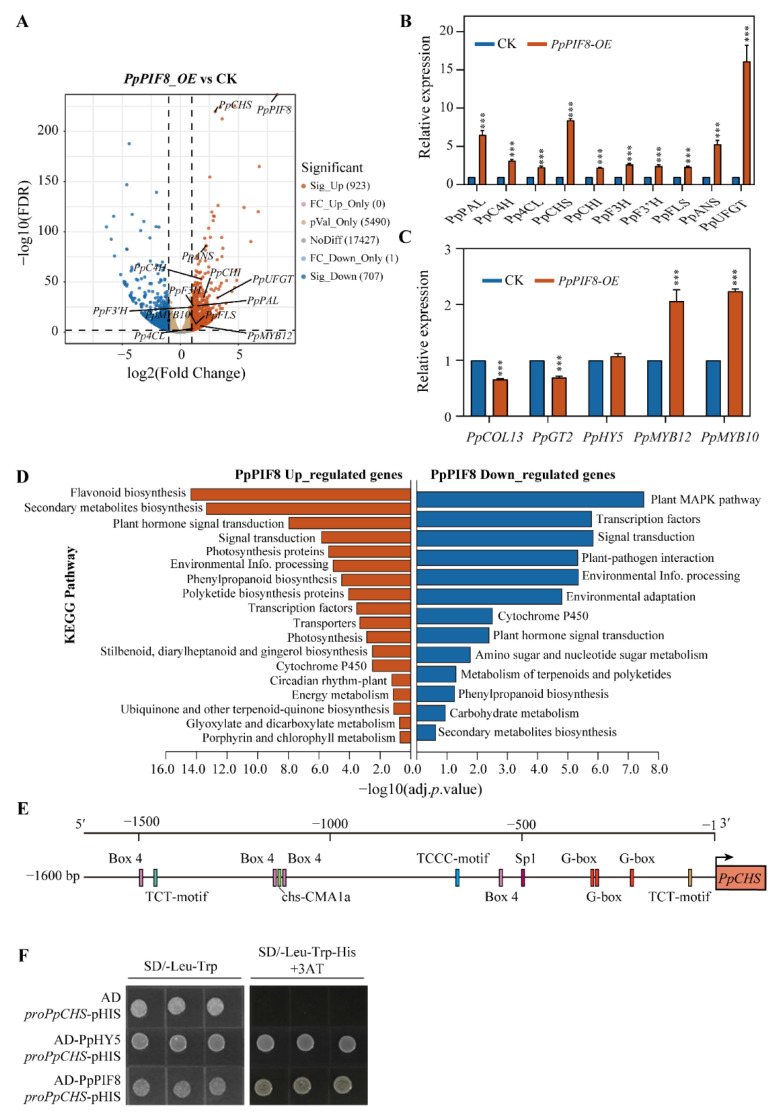
RNA-Seq analysis of *PpPIF8_OE* transgenic pear calli. (**A**) A volcano plot illustrating DEGs between *PpPIF8_OE* and control pear calli. Genes significantly upregulated and downregulated in *PpPIF8_OE* pear calli are shown in red and blue, respectively. (**B**) Validation of anthocyanin biosynthesis genes expression by RT-qPCR. Error bars denote the standard deviation. Asterisks above the bars indicate significant differences (*** *p* < 0.001) obtained by two-tailed Student’s *t*-test. (**C**) Validation of transcription factor expression by RT-qPCR. Error bars denote the standard deviation. Asterisks above the bars indicate significant differences (*** *p* < 0.001) obtained by two-tailed Student’s *t*-test. (**D**) KEGG pathway analysis of upregulated and downregulated genes in *PpPIF8_OE* pear calli. (**E**) Promoter analysis of *PpCHS*. (**F**) Yeast one-hybrid analysis of the interaction between PpPIF8 and the *PpCHS* promoter. SD/−Leu−Trp, synthetic dextrose medium lacking Leu and Trp; SD/−Leu−Trp−His+3AT, synthetic dextrose medium lacking Leu, Trp, and His plus 5 mM 3AT.

**Table 1 ijms-23-04798-t001:** Transcription factors in “Blue” and “Turquoise” modules.

Type of TF	No.	Description
AP2-ERF	35	Ethylene responsive transcription factor
AUX/IAA	6	AUX/IAA transcriptional regulator
bHLH	18	Basic helix–loop–helix protein
bZIP	7	Basic-leucine zipper protein
B-box	8	B-box type zinc finger family protein
C2H2	12	C2H2-type zinc finger family protein
C3H	3	zinc finger (CCCH-type) family protein
GNAT	7	Acyl-CoA N-acyltransferases protein
GRAS	5	GRAS family transcription factor
HB	14	Homeobox-leucine zipper protein
LOB	5	LOB domain-containing protein
MYB	40	Myb domain protein
NAC	15	NAC domain-containing protein
SWI/SNF	3	SWIB/MDM2 domain superfamily
TAZ	3	BTB and TAZ domain protein
Tify	3	Jasmonate-zim-domain protein
Trihelix	7	Trihelix transcription factor
WRKY	24	WRKY family transcription factor
Other TFs	48	
Total TFs	263	

**Table 2 ijms-23-04798-t002:** Characteristics of phytochrome-interacting factors (PIFs) genes in pear.

Gene Name	Gene ID	Amino Acid Length (aa)	MW (KD)	pI	Instability Index	Best Hits
*PpPIF1*	gene2064 (LOC103961475)	508	55.21	9.39	68.07	AtPIF1
*PpPIF3*	gene16013 (LOC103955840)	713	76.40	6.09	53.64	AtPIF3
*PpPIF3a*	gene15524 (LOC103955304)	716	76.56	6.12	55.57	AtPIF3
*PpPIF5*	gene8334 (LOC103947396)	550	60.37	6.47	54.50	AtPIF5
*PpPIF5a*	gene2521 (LOC103965596)	541	59.29	6.32	57.78	AtPIF5
*PpPIF8*	gene35851 (LOC103935970)	442	47.55	7.14	54.29	AtPIF8
*PpPIF7a*	gene31629 (LOC103931367)	398	44.00	9.20	71.73	AtPIF8
*PpPIF7b*	gene3216 (LOC103931008)	406	44.85	9.02	64.49	AtPIF8

## Data Availability

The raw RNA-Seq data of this study have been deposited in the Genome Sequence Archive in China National Center for Bioinformation (https://www.cncb.ac.cn/, accessed on 1 April 2022) under the project number: PRJCA009255.

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
