# Peer review of "RNA-Seq Analysis Identifies Transcription Factors Involved in Anthocyanin Biosynthesis of ‘Red Zaosu’ Pear Peel and Functional Study of PpPIF8"

_ijms, 2022, doi:10.3390/ijms23094798_

Round 1

Reviewer 1 Report

This work presents bioinformatics analysis of anthocyanin biosynthesis in pear fruit. The authors studied transcription factors in this plant using gene expression network models. PpPIF8 gene was considered in details based on combination of experimental techniques.

I have some minor remarks.

First, it is worthy to mention standard plant name in the title and in the abstract.

Wording ‘pear skin’ is not enough. Just add word ‘pear (Pyros)’ to indicate the plant species. Maybe even indicate ‘Red Zaosu’ pear cultivar to be precise. Now the description of the plant and cultivar is given only in Methods section. It is not clear from the beginning what is the species studied. The results found may depend on the pear cultivar.

Line 17: ‘in red pear’ - add Latin name of the species, or name ‘Red Zaosu’ to this phrase.

Line 18: ‘263... factors’ - it is too large number, might be half of all the transcription factors in pear genome. Please add % of total number, or try to rephrase. Moreover ‘gene modules (Turquoise and Blue..)’ looks not clear from the beginning. These modules’ names just were given by the authors. I suggest remove it from the Abstract, and comment later in the text, giving it in parentheses ‘Turquoise’ and ‘Blue’.

Line 19029: ‘STRING and CetnriMo’ - need comments what it is, give full names, or comment that it is gene network modeling tools. ‘STRING’ should be ‘STRING-DB’. Need give references, version number and web-links to these tools (in the text, not in the ABstract)

Line 29: Keywords - should be extended - add words ‘Pyrus’, ‘transcription factors’, give WGCNA abbreviation in full.

Line 35: ‘anthocyanin has a preventive effect on cancer’ - this idea should be extended by additional references. There are some works on treatment effects on anthocyanin in other plants, crops. Please add some text and references here. Reference [2] only is not enough.
Line 51: ‘Loss-function of HY5 reduces the anthocyanin level [6-8].’ - need rephrase - add word ‘experiments’, cite 6, 7 and 8 references separately. it is kind of redundant (bulk) citation.

Line 59: [10-12] - cite less, or cite these references separately.

Line 71: [18-21] - bulk citation. Give these references in the phrase for each fruit separately, change the phrase.

Line 93: ‘pear’ - need comment on the cultivar and reference pear genome used in the study (briefly from the Methods section). It is not clear from the beginning - used Pyrus species, control species. Mention Pyrus bretschneideri ‘DangshanSuli’ pear as the reference, and ‘Red Zaosu’ as the object of the study. I see it only later in the text. A reader from general bioinformatics background may not understand the terminology used.

Line 94: ‘CentriMo analysis’ - what is CentriMo? - need add comments, a reference to this tool.

Line 105: ‘at 120 DAFB’ - this is not clear, need give abbreviation DAFB in full.

Line 129: ‘We performed PCA and sample correlation analysis..’ - try to rephrase, avoid word ‘performed’

Line 152: ‘were adopted’ - please rephrase. ‘Adopted’ is not appropriate word. Just write like ‘We used the following threshold..’

Mention p-value threshold used in this work.

Line 174: ‘PpCHS,  PpFSL’ - what is known about these genes’ function? Maybe any known function in other plant species?

Lint 184: ‘WGCNA analysis’ - need give WGCNA abbreviation in the section title in full.

Currently it is given only in the text in line 188.

Line 189-190: “Turquoise” module ... “Blue” module - add some comments why such names were used? Is it arbitrary ‘color’ names? Add some phrase, not to mix with module names with other studies. Always use parentheses for such names.

Line 210: ‘performed the enriched ... (KEGG)’ - need add a reference. Change the phrase. Not use ‘perform’ word. Write, for example, ‘We analyzed gene ontology terms enrichment using ...’

Line 218: ‘Transcription factors (TFs) play a vital role in regulating anthocyanin biosynthesis. ‘ - need add a reference to this statement.

Line 220: ‘iTAK’ - need give abbreviation in full, comment what this analysis is about.

Line 254: ‘STRING analysis’ - add the reference to STRING-DB, cite it correctly. From the panel (C) it looks not like standards STRING network visualization. Comment about additional tools used, if any.

Line 255: ‘...in “Blue” and “Turquoise” modules’ - avoid using these names in the section title. Or write more correct phrase ‘...in the defined “Blue” and “Turquoise” modules’

Line 271: ‘CentriMo analysis’ - add a reference

Line 274: ‘PIFs’ - give the abbreviation in full, or write ‘PIF family genes’

Line 323: ‘Pearson’  - not need Italic font here.

Line 334: ‘We cloned the full-length CDS...’ - this is complex experiments - need add a reference, cite the protocol used for this cloning work.

Line 360: ‘PpPIF8_OE’ - this is new abbreviation - need comment on it. Is it new clone with the overexpressed gene?

Line 374: ‘subjected to RNA-Seq analysis’ - please rephrase. It is kind of science jargon ‘subjected to’.

Line 387: ‘KEGG enrichment analysis was performed’ - rephrase this sentence. Avoid wording ‘was performed’, add ‘gene ontology’ words.

Lines 389 and 391 - remove ‘etc.’ word. It is redundant in the same sentence.

  1. Materials and Methods section only starts from page 20.

So the fruit name ‘Red Zaosu’ (Pyrus pyrifolia× Pyrus communis) and DAFB abbreviation appeared only here. It should be mentioned in the beginning to be understable by the readers. At least in short form.

Line 618: ‘to blast’ - it is also kind of jargon. Try to rephrase like ‘we search homologs by BLAST using...’

TAIR abbreviation could be given in full, with the release number.

Line 626: ‘MUSCLE’ - may give abbreviation in full, web-link to this tool.

Line 668: ‘between B and D group samples’ - may remove this words from the phrase for clarity. B and D groups are not related to the test used.

Line 671: ‘This section is not mandatory but can be added..’  - remove this text, evidently it is a typo.

Line 684: ‘Funding:Please add:’ - remove words ‘Please add’

Lines 687-688 ‘Not applicable.’ - mat change to ‘N/A’ or remove.

But for line 689: ‘Data Availability Statement’ it is worthy add that the data are available in the Supplement. Or deposit RNA-seq data to some resource. Indicate the experimental data availability, please.

Reviewer 2 Report

General comments:

This is a comprehensive piece of work performed to a good standard and worthy of publication in IJMS. I do have some suggestions below that I think will improve the paper and place it into the context of what is known already about the PIF8 gene (in other Rosaceae in particular). I suggest they focus more on the considerable body of data they have added to what was previously known, rather than imply (largely by omission admittedly) that this may be the first to discovery of the role of PpPIF8 in light controlled regulation of anthocyanins. I suggest they do a more comprehensive job reviewing what is known about this branch already in other plants, and in Rosaceae in particular.

Most of my other comments are minor, cosmetic/improvements/minor grammar corrections and are given below.

Title: I find the title too long suggest they trim it a little

Abstract: The authors insight into PpPIF8 is not quite as unique as the abstract indirectly implies (largely by omission) as this has already been identified in other closely related Rosaceae the authors need to acknowledge this and perhaps focus a bit more on what is different or what they have added to the other Rosaceae data.

Introduction:

PIF7 and PIF8 studies in Arabidopsis suggest they play important roles in light response –as they are related to the PIF8 cluster and mentioned in the discussion the authors should mention some of this literature and any relevant MdPIF7/8 literature in their introduction

Results:

Corrections

L140 ....showed strikingly differences...

L247-249 awkwardly worded sentence that is trying to get a key concept across, suggest rewording it without “besides” which leads to uncertainty if they are referring to something “other” than PpPIF8 when I think they don’t wish to imply this...

Line 160 statement a little too absolute as there will still be some rare genes in the network linked to deactivation of gene expression-suggest change to; ..debagging is largely due to activating.....

L260: I am not clear if they are referring to 87 TFs in the gene set or 87 TF binding sites upstream of the gene set they analysed-I suspect it is the latter as they go on to discuss this later in the paragraph so please clarify in L260...

L285: Isolated protein stability may not be that significant –in protein complexes this instability may well be “compensated”, does this instability index link with the degree of disorder found in these TFs-that would be interesting to identify as that may link with their role as hub proteins (eg see Sun et al. 2013 Plant Cell 25: 38)?

L389 & 393a phrase like etc does not seem appropriate in a scientific publication as it is completely vague..!

L394 delete “Since” as its small FDR value will not have been directly caused by being a “key enzyme” you could add ”however” after the comma instead to more accurately reflec what I think you are trying to say...

L401-402 awkwardly worded sentence suggest you end sentence at Figure 11E and start a new sentence “The G-box is known to be a PIF binding...why was yeast 1 hybrid with deletion of the G-boxes in the promoter not tried???

Discussion

L427 ..of fruit peel due to blocking light penetration.

L456 ..PpPIF8 clustering was highly

L467...PpPIF8 transcripts were...

L499-507 Could another possible reason be the nature of the anthocyanin mutations in these two genotypes-does Yunhongyihao ie have the same MdBBX24 mutation-if not could this possibly be the explanation-suggest the authors elaborate on reasons why/why not this could be responsible...?

Materials and Methods

Line 561: it seems that the OE vector must have a GFP fusion of some type as GFP antibody is used to detect the protein in Fig 10C. CAN the authors please clarify if their construct or one of their vectors contains a GFP fusion protein in this section.

I assume that Pyrus communis ‘Xiuzhenxiang’ does not show red colouration nor have any particular anthocyanin inducing alleles?  It would be nice to confirm its anthocyanin accumulation status in the M&M to place the RNA-seq performed with it in context-in particular if PpBBX24 shows anything interesting, presumably in its “fully functional” form...

How does the PpBBX24 mutation in their sampling strategy potentially impact on their results as it is clearly an interacting partner of PpPIF8? I would like to see some discussion of this as their findings could potentially be quite dependent on their “PpBBX24 minus” Red Zoasu background

Conclusions:

Attention to detail-they have not included this section but left a comment that is clearly derived from the instructions (L671-672)

Figures

6A the figure legend should make it clear this is the 55 genes from the Blue and turquoise modules referred to in the text

8B given the high degree of synteny between apple and pear it would be interesting to know if the two sets-MdPIF8 & PpPIF8, and MdPIF7 & PpPIF8a/b are located in syntenic regions of apple and pear and if the PIF7 and PIF8 “modules” (based on their Malus names)  are located on homeologues based on the well known paleopolypoid doubling of the Maloideae –if so I suggest renaming PpPIF8a and PpPIF8b as PpPIF7a and PpPIF7b instead...

8C authors need to explain what D and B means in this figure –I can see from the methods it is bagging (B) and debagging (D) but should be able to deduce this based on the legend alone.
